# Molecular Signatures of Tumour and Its Microenvironment for Precise Quantitative Diagnosis of Oral Squamous Cell Carcinoma: An International Multi-Cohort Diagnostic Validation Study

**DOI:** 10.3390/cancers14061389

**Published:** 2022-03-09

**Authors:** Muy-Teck Teh, Hong Ma, Ying-Ying Liang, Monica Charlotte Solomon, Akhilanand Chaurasia, Ranjitkumar Patil, Satyajit Ashok Tekade, Deepika Mishra, Fatima Qadir, Ji-Yun Stephanie Yeung, Xinting Liu, Safa Kriuar, Ruoqi Zhao, Ahmad Waseem, Iain L. Hutchison

**Affiliations:** 1Centre for Oral Immunobiology & Regenerative Medicine, Institute of Dentistry, Barts & The London School of Medicine and Dentistry, Queen Mary University of London, London E1 2AT, UK; f.qadir@qmul.ac.uk (F.Q.); ji-yun.yeung1@nhs.net (J.-Y.S.Y.); xinting.liu1@nhs.net (X.L.); safa.kriuar@kcl.ac.uk (S.K.); zhaorq19@fudan.edu.cn (R.Z.); a.waseem@qmul.ac.uk (A.W.); iain.hutchison@savingfaces.co.uk (I.L.H.); 2China-British Joint Molecular Head and Neck Cancer Research Laboratory, Department of Oral & Maxillofacial Surgery, Affiliated Hospital & School of Stomatology, Guizhou Medical University, Guiyang 550004, China; mahong@gmc.edu.cn; 3Department of Radiation Oncology, Affiliated Cancer Hospital & Institute of Guangzhou Medical University, Guangzhou 510000, China; yingyingliang@gzhmu.edu.cn; 4Department of Oral Pathology & Microbiology, Manipal College of Dental Sciences, Manipal, Manipal Academy of Higher Education, Manipal 576104, Karnataka, India; monica.charlotte@manipal.edu; 5Department of Oral Medicine & Radiology, Faculty of Dental Sciences, King George’s Medical University, Lucknow 226003, Uttar Pradesh, India; akhilanandchaurasia@kgmcindia.edu (A.C.); ranjitkumarpatil@kgmcindia.edu (R.P.); 6Department of Oral & Maxillofacial Pathology, Modern Dental College & Research Centre, Indore 453112, Madhya Pradesh, India; satyajitraje@gmail.com; 7Department of Oral Pathology & Microbiology, Centre of Dental Education & Research, All India Institute of Medical Sciences, New Delhi 110029, India; deepikamishra@aiims.edu; 8Research Division, Saving Faces—The Facial Surgery Research Foundation, London WC1H 9DZ, UK; 9Research Division, The National Facial, Oral and Oculoplastic Research Centre, London WC1H 9DZ, UK

**Keywords:** molecular diagnostics, early oral cancer biomarkers, qMIDS, squamous cell carcinoma, FOXM1 diagnostic biomarkers, microenvironment, prognostic biomarkers, clinical translation, personalised medicine, early detection, oral premalignant disorders, dysplasia

## Abstract

**Simple Summary:**

Incisional tissue biopsy is highly invasive, but it remains the current practice for oral cancer diagnosis through histopathological interpretation. To reduce biopsy invasiveness without compromising diagnosis, we pioneered a multigene RT-qPCR method for cancer detection using only a tiny 1 mm^3^ minimally invasive biopsy. Here we presented international multicohort validation of our second-generation method which involved new genes from matrix/stroma and immune regulations enabling sensitive, quantitative and precise oral cancer detection in otherwise ambiguous oral lesions.

**Abstract:**

Background: Heterogeneity in oral potentially malignant disorder (OPMD) poses a problem for accurate prognosis that impacts on treatment strategy and patient outcome. A holistic assessment based on gene expression signatures from both the tumour cells and their microenvironment is necessary to provide a more precise prognostic assessment than just tumour cell signatures alone. Methods: We reformulated our previously established multigene qPCR test, quantitative Malignancy Index Diagnostic System (qMIDS) with new genes involved in matrix/stroma and immune modulation of the tumour microenvironment. An algorithm calculates and converts a panel of 16 gene mRNA expression levels into a qMIDS index to quantify risk of malignancy for each sample. Results: The new qMIDS^V2^ assay was validated in a UK oral squamous cell carcinoma (OSCC) cohort (*n* = 282) of margin and tumour core samples demonstrating significantly better diagnostic performance (AUC = 0.945) compared to previous qMIDS^V1^ (AUC = 0.759). Performance of qMIDS^V2^ were independently validated in Chinese (*n* = 35; AUC = 0.928) and Indian (*n* = 95; AUC = 0.932) OSCC cohorts. Further, 5-year retrospective analysis on an Indian dysplastic lesion cohort (*n* = 30) showed that qMIDS^V2^ was able to significantly differentiate between lesions without transformation and those with malignant transformation. Conclusions: This study validated a novel multi-gene qPCR test on a total of 535 tissue specimens from UK, China and India, demonstrating a rapid minimally invasive method that has a potential application for dysplasia risk stratification. Further study is required to establish if qMIDS^V2^ could be used to improve OPMD patient management, guide treatment strategy and reduce oral cancer burden.

## 1. Introduction

The global cancer statistics 2020 (GLOBOCAN 2020) indicated oral cancer (including lip, C00–C06) as one of the eight leading cancer types for both incidence and mortality in men [1]. Although there have been many advances in surgery, reconstruction [2], full dental rehabilitation and psychological support leading to improved survival and better quality of life [3], there is still a great need to reduce the morbidity and mortality of this most socially disabling cancer by improving early detection and treatment.

Oral squamous cell carcinoma (OSCC) represents a large proportion of head and neck cancers. Majority of OSCC patients have some form of pre-existing oral potentially malignant disorder (OPMD) lesions amenable to early diagnosis and risk stratification [4,5,6,7,8]. OPMDs are very common with overall prevalence ranging from 10.54% in Asia, 3.94% in South America, 3.72% in Middle East, 3.07% in Europe and 0.11% in North America [9]. Although over 70% of OSCC are preceded by OPMDs [4,5,10], pathologists are sometimes not in agreement as histopathological grading of epithelial dysplasia is complex [11], time-consuming, subjective and susceptible to misdiagnosis due to sampling error [6,7,12,13]. Although a number of clinical trials have investigated non-surgical interventions for treating OPMDs to prevent malignant transformation, none have shown significant clinical responses [14,15,16,17]. At present there is currently no quantitative method in widespread use for OPMD malignant transformation risk assessment. Therefore, most if not all OPMD patients are either put on regular surveillance or are biopsied and put on a variable period of review before discharge [4,5,6,8,12]. Hence, for the low-risk OPMD patients, regular review will result in unnecessary health service costs and may cause continued anxiety for the patients. For the high-risk patients (<12%) [10], early discharge from review will probably result in late presentation of OSCC resulting in increase in morbidity and treatment costs and increased risk of death.

A systematic review estimated that OPMD has a 12% mean risk of malignant progression [10]. In the UK, it was found that only 7% of OPMD patients, referred using the two-week ‘fast-track’ suspected cancer referral system, actually developed OSCC [18]. An audit (2002–2012) carried out in a UK district general hospital found a significant 450% increase in the annual number of ‘fast-track’ referral patients but cancer detection rate decreased by 50% [19] indicating an unnecessary burden on secondary care due to large number of false positives being referred. Another UK National Head and Neck Cancer Audit (2014) found that over 70% of patients waited between 10 and 21 days from biopsy to pathology reporting. It is known that delayed treatment directly causes poor long-term morbidity and survival [4,6,20,21]. Cost-effectiveness studies in UK and Taiwan have independently shown significant cost savings when OSCC patients were treated at early stages (premalignant or stage 1) compared to late stages 2–4 [22,23]. Collectively, these evidenced a huge burden to the healthcare systems. It is clear that early OSCC diagnosis is the key to improving patient outcome [4,6,20,21,22,23]. Therefore, a rapid cost-effective diagnostic method that can identify high-risk OPMD patients would help alleviate the current disease and financial burdens for treating OPMD and OSCC patients.

With a multi-gene expression qPCR test such as the quantitative Malignancy Index Diagnostic System (qMIDS) which requires only 1 mm^3^ tissues for diagnosis, we have previously demonstrated utility for quantitative OSCC diagnosis and for OSCC surgical margin analysis [24]. Although OPMDs may exhibit different molecular signature to that found in tumour, it is known that high-risk OPMDs exhibit chromosomal and genomic instability alterations indicative of malignant transformation [25]. As chromosomal, epigenomic and genomic perturbations would likely lead to altered transcriptome profile, utilising a panel of gene expression signature as surrogate disease markers as in qMIDS [24] or others [26,27,28,29] has shown to be effective for predicting risk of early malignant transformation in OPMDs.

Over the course of development and validation of the qMIDS test for early OSCC diagnosis and prognosis, we previously tested 1761 individual 1-mm^3^ tissue specimens collected from 427 patients represented by Caucasians, South Asians and East Asians [24,30]. As the qMIDS test involves measuring 16 genes (14 target + 2 reference) in each sample, this generated a large resource of gene expression data (24,654 data points) from clinical samples. Although all 14 target genes were originally found to be differentially expressed between normal and cancer cell lines [24], a closer look in our qMIDS clinical sample dataset showed that some genes turned out to be less differentially expressed in tissue samples compared to cell lines. This study investigated the possibility to evolve and improve our original qMIDS^V1^ assay [24,30] by including new genes with functions in stroma/matrix and immune regulation to enhance diagnostic performance for early oral cancer detection and/or to predict malignant transformation in OPMDs. 

## 2. Materials and Methods

### 2.1. Clinical Samples

The use of human tissue was approved by the relevant Research Ethics Committees at each institution: Queen Mary University of London (UK NREC:06/MRE03/69 and QMERC20.142), Kasturba Hospital, Manipal (IEC 343/2017), King George’s Medical University, Lucknow, India (104th ECM IIA/P18), Affiliated Cancer Hospital & Institute of Guangzhou Medical University, Guangzhou, China (GMU32/2019). All tissue samples were collected according to local ethical committee-approved protocols and informed patient consent was obtained from all participants. Clinico-histopathological reports of the tissue samples were obtained from collaborating clinicians and pathologists at each institution. Inclusion and exclusion criteria for all cohorts: all available tissue surplus to diagnosis at each institution that were clinicopathologically confirmed to be normal oral mucosa, oral lichen planus, oral leukoplakia, oral submucous fibrosis and sporadic oral squamous cell carcinomas were included except the following exclusion criteria: 1, samples that failed to provide sufficient quality RNA to enable SYBR green fluorescent detection of 2 reference genes (YAP1 and POLR2A) with cycle threshold >35 by RT-qPCR quantification (see below). In the UK cohort, only margin samples with histologically normal epithelium were included.

Dysplasia cases were graded according to the World Health Organization (WHO) 2017 OED grading criteria into mild, moderate or severe (reviewed in [11]). Of the 70 dysplasia cases, 30 cases (2010–2016) had clinical follow-up (of at least 12 months) outcome data whereby patients that did not develop OSCC were classified as non-transformed and patients that developed OSCC within a 5-year period were classified as transformed (Table 1).

For the UK and China cohorts, fresh oral tissues were preserved in RNALater (#AM7022, Ambion, Applied Biosystems, Warrington, UK) and stored short-term at 4 °C (1–7 days) prior to subsequent storage at −20 °C until mRNA extraction (Dynabeads mRNA Direct kit, Invitrogen, ThermoFisher Scientific, Paisley, UK) [24,30]. In the Indian cohort, FFPE samples were each (2–8 curls of 10 µm thick sections) deparaffinised with xylene (1 mL, repeated once) followed by rehydration (1 mL each of 100%, 90% then 70% ethanol) prior to air drying (60 °C, 5 min) followed by total RNA purification (Qiagen FFPE RNeasy Kit, #73504). All samples were coded, link-anonymised and tested blindly to ensure that the qMIDS assays were performed objectively. All patient demographic distribution data (age, sex, substance use habits, and anatomical sites of lesion) associated with tissue samples used in this study are summarised in Figure 1 and Table 1. All digital clinicopathological records were stored in password-protected computer files.

### 2.2. The qMIDS Assay

The qMIDS assay methodology was performed as described previously [24,30] with minor modifications. Briefly, in the qMIDS assay, the present assay format involves using either qPCRBIO SyGreen 1-Step Go (PCR Biosystems, London, UK, PB25.31-12) for one-step using RNA as input material or qPCRBIO SyGreen Blue Mix Lo-ROX (PCR Biosystems, PB20.15-51) for two-step using cDNA as the input material. Relative quantification of 14 target genes and 2 reference genes were performed using a 384-well format LightCycler 480 qPCR system (Roche, Basel, Switzerland) based on our previously published protocols [24,31,32,33], which are MIQE compliant [34]. Briefly, thermocycling begins with 45 °C for 10 min (for reverse transcription—this step is omitted when using the 2-step reagent with cDNA as input material) followed by 95 °C for 30 s prior to 45 cycles of amplification at 95 °C for 1 s, 60 °C for 1 s, 72 °C for 1 s, 78 °C for 1 s (data acquisition). To maximise primer specificity, we included a ‘touch-down’ annealing temperature intervention (prior to the amplification step) with a starting temperature at 66 °C followed by a stepwise reduction of 0.6 °C/cycle; 8 cycles. After the amplification step, melting analysis (95 °C for 30 s, 75 °C for 30 s, 75–99 °C at a ramp rate of 0.57 °C/s) was performed to validate single product amplification in each well (See Appendix A). A second derivative maximum algorithm [35] (Roche) was used to calculate the relative ratios of mRNA transcripts (target:reference). All primers for qMIDS^V1^ were published previously [24], whereas, qMIDS^V2^ primers are provided in Appendix A. Each target gene was normalised to both YAP1 and POLR2A reference genes, as validated previously to be the most stable reference genes across a wide range of human primary epithelial keratinocytes, dysplastic and HNSCC cell lines [31], using the GeNorm algorithm [36]. The qMIDS^V1^ vs qMIDS^V2^ workflow and detail 384-well qPCR assay methods are provided in Appendix A. Relative expression data were then exported into Microsoft Excel for computing qMIDS scores based on our previously published qMIDS algorithm [24]. No template controls (NTC) were prepared by omitting tissue sample during RNA purification and eluates were used as NTCs for qMIDS assay. 

### 2.3. Statistical Analysis

Scatter plots were analysed using polynomial regression (y = a + b_1_x + b_2_x^2^ + b_3_x^3^) on both raw and Log_2_ ratio data of each target gene to survey its correlation with qMIDS values. Statistical *t*-tests *p* values were used for differential analysis between two groups of data. Mann–Whitney U nonparametric test was also used to test between two groups of data with skewed distribution. Fisher’s exact test and Pearson’s chi-square test were used for multi-dimension small sample size datasets. Diagnostic test performance comparison data were calculated using a Diagnostic Test Calculator [37]. Receiver operating characteristics (ROC) curves were generated from qMIDS data to obtain area under the ROC curves (AUC) to assess assay diagnostic performance. Beeswarm Boxplots were created in R (version 2.13.1; The R Foundation for Statistical Computing) [38].

## 3. Results

### 3.1. Gene Selection

Since our first publication validating the use of qMIDS for early OSCC diagnosis [24], we have accumulated large number of qMIDS data on 1761 tissue fragment samples (each with relative gene expression values of 14 target genes) from normal and disease (OPMD or OSCC) tissue specimens donated by patients from UK and Norway, totalling to 24,654 gene expression data points. Over the course of optimising the qMIDS assay for OSCC diagnosis, we noticed that some target genes were less contributory which may have compromised the performance of the original qMIDS^V1^ test’s ability to discriminate between low and high risk OPMDs. Therefore, using our previous qMIDS^V1^ (RT-qPCR) data generated from clinical samples as a training dataset, we aimed to remove less significant differentially expressed genes from the qMIDS gene panel. We subjected our qMIDS^V1^ data to two methods of analyses: 1. Distribution with correlation regression analysis, and, 2. threshold (cut-off at 4.0 [24]) methods. For the distribution method, we first performed a correlation regression analysis between each gene with qMIDS^V1^ index value for each of the *n* = 1761 samples, generating scattered dot-plots with regression analysis (Appendix A). We then subjected our dataset to three methods of sub-groupings (following equal, skewed or Gaussian distributions) prior to linear and polynomial curve-fitting methods to access how well each gene correlated with qMIDS^V1^ index values (Appendix A). For the threshold method, we segregated samples into normal (*n* = 1189) vs. disease (*n* = 572) based on our previously determined cut-off value at 4.0 [24]. The Student’s *t*-test was performed on each of the 14 target genes (Appendix A, beeswarm plots on right panels) where *p* values and fold changes are shown in Appendix A. Final gene arbitrary scores were calculated from both methods whereby 6 genes (HOXA7, CENPA, NEK2, DNMT1, FOXM1, IVL) scored >7 and 8 genes (MAPK8, CCNB1, AURKA, CEP55, BMI1, HELLS, DNMT3B, ITGB1) scored below 7 (Figure 2A; Appendix A).

### 3.2. Reformulate Gene Panel for qMIDS Assay

To reduce the number of biomarkers measured in the qMIDS gene panel, we tested if a gene panel of 12, 10, 8 or 6 (instead of 14) genes could maintain the qMIDS diagnostic accuracy and sensitivity. Unfortunately, reducing from 14 to 12, 10, 8 or 6 genes rendered the qMIDS test results progressively inferior and unreliable (data not shown). Hence, to maintain diagnostic performance and assay consistency with our previously validated qMIDS assay format [24,30] (see Appendix A), we opted for replacing those 8 less influential genes (with score <7) by adding back 8 new candidate genes (selected through literature review and Oncomine^TM^/GEO database searches) with functional implications in stromal matrix and immune modulation in squamous cell carcinomas. A new panel of candidate genes (~20) was first shortlisted according to methods previously described [24,30,39] and each gene was individually tested (using RT-qPCR) for their significance of differentiating normal from cancer cell lines and tissue samples (data not shown). Eventually, we selected 8 most significant genes (INHBA, TOP2A, BIRC5, MMP13, CXCL8, NR3C1, CBX7, S100A16) to be combined with 6 previous genes (HOXA7, CENPA, NEK2, DNMT1, FOXM1, IVL) from qMIDS^V1^ to form a new gene panel in qMIDS^V2^ (Figure 2A).

### 3.3. Comparison between qMIDS^V1^ and qMIDS^V2^

We hypothesised that by removing these less influential genes and replacing them with new genes involved in stroma/matrix and immune modulation would improve qMIDS test performance for detecting those OPMDs with early malignant gene signature and so most likely to transform into OSCC. Due to heterogeneity of tumour tissue samples, we first performed comparison between qMIDS^V1^ vs. qMIDS^V2^ on a case study using a T3 OSCC tumour core sample (UK). We cut this tissue into 10 pieces of ~1 mm^3^ fragments (Figure 2B) and cDNA was generated from each tissue fragment for analysis with both qMIDS^V1^ and qMIDS^V2^ assays simultaneously using a 384-well format qPCR system (Appendix A). qMIDS^V1^ generated lower index values in most of the tissue fragments compared to qMIDS^V2^. Collectively, the median/mean values for qMIDS^V1^ vs. qMIDS^V2^ were 5.0/6.2 vs. 7.7/8.9, respectively (Figure 2C), which were statistically different (*p* < 0.0001). This indicates that qMIDS^V2^ may be more sensitive than qMIDS^V1^. According to the clinicopathological data, this specimen was a T3 tumour. Therefore, a qMIDS index value of 7.7–8.9 would be more appropriate than 5–6.2, given that normal-disease cut-off value was 4.0 and malignancy index value of tumours was up to 14.0 [24]. 

To test if qMIDS^V2^ has superior discriminatory power between margin and tumour over qMIDS^V1^, we have chosen two UK cohorts of patients which were previously tested and failed to be segregated by qMIDS^V1^. The first cohort contains paired margin-tumour core samples from the same patients (*n* = 7), the second cohort consisted of margin samples (*n* = 5) and tumour core samples (*n* = 5) from different patients. We have previously shown that measuring multiple sub-fragments from a single biopsy increases the diagnostic accuracy due to the ability to topologically map tumour heterogeneity [24]. Hence, each tissue sample was cut into 9 to 24 pieces (depending on the size of biopsy) of about 1 mm^3^ each sub-fragment. A total of *n* = 498 sub-fragments (from paired samples of 7 patients) and *n* = 204 sub-fragments (unpaired samples of 10 patients) were independently analysed for qMIDS^V1^ vs. qMIDS^V2^ test comparison on each fragment (Figure 2D). As per our original findings, our current data confirmed that qMIDS^V1^ failed to differentiate between margin and core tumour samples but qMIDS^V2^ significantly discriminated between tumour margins and tumour core samples (Figure 2D). We concluded that for both cohorts of paired and unpaired samples, qMIDS^V2^ outperformed qMIDS^V1^ in segregating margin from core tissue samples. 

### 3.4. Validation in UK OSCC Cohort

To further validate qMIDS^V2^, we tested qMIDS^V2^ on a UK OSCC cohort (*n* = 282) and compared with a subset of samples tested by qMIDS^V1^ (*n* = 102; Figure 3). The unequal sample size was due to insufficient tissue samples left for analysis. In agreement with the above case studies (Figure 2B–D), qMIDS^V2^ showed overall superior diagnostic performance compared to qMIDS^V1^. Most notable was the increase in sensitivity/specificity/accuracy from 72%/71%/72% in qMIDS^V1^ to 88%/96%/92% in qMIDS^V2^ (Figure 3C). Importantly, the false positive rate was reduced from 29% to 4.5% and false negative rate was reduced from 28% to 12% in the qMIDS^V2^. These data confirmed that our strategy of removing less influential genes based on large gene expression datasets (24,654 data points) obtained from clinical tissue samples and by including genetic signatures of the tumour microenvironment (stroma/matrix/immune regulations) in addition to the genetic signature of tumour cells, could significantly improve qMIDS diagnostic performance to enable highly precise quantitative diagnosis of OSCC. 

To simplify and economise the assay, we attempted to reduce the number of genes in the qMIDS^V2^ assay. We tested a reduced 10-gene panel (qMIDS^V2^* by deleting 4 less effective target genes from the 14-gene panel qMIDS^V2^; Appendix A). We also performed sequential removal of individual genes from the qMIDS^V2^ assay to see how each gene contributed to the overall diagnostic performance (Appendix A) in the UK cohort. We found that the full 14-gene panel provided the best overall diagnostic performance. Removing any single gene from the qMIDS^V2^ negatively affected diagnostic performance (Appendix A). This confirmed that the full 14 target gene panel (plus two reference genes) in the qMIDS^V2^ assay was an optimal minimal combination of biomarkers that produce the best diagnostic performance in our study.

### 3.5. Validation in Chinese Oral SCC Cohort

The previous qMIDS^V1^ had been previously validated in different ethnicities (UK, Norway [24] and China [30]) showing comparable results. However, due to change in biomarker panel in qMIDS^V2^, here we re-validated qMIDS^V2^ on a Chinese patient cohort with *n* = 35 individuals contributed fresh frozen tissues whereby 11 were normal oral mucosa (NOM) and 24 OSCC tissues. The qMIDS^V2^ assay segregated significantly between NOM (median/mean: 0.5/1.2 ± 1.8) and OSCC (6.7/6.6 ± 2.7, *p* < 1 × 10^−6^; Figure 4A).

### 3.6. Validation in Indian Oral SCC Cohorts

To further rule out possible confounding variables associated with ethnicity and geographical differences, we validated the qMIDS^V2^ on an Indian cohort consisted of a total of *n* = 218 FFPE archival samples of which 35 were normal oral mucosa, 70 dysplasias, 60 OSCC, 37 oral submucous fibrosis (OSF) and 16 oral lichen planus (OLP). Within the Indian cohort, there was significant segregation between NOM (median/mean ± SD: 1.4/1.8 ± 2.5) and mild/moderate dysplasias (5.6/5.1 ± 2.9, *p* < 7 × 10^−7^), severe dysplasias (4.7/4.3 ± 2.6, *p* < 1 × 10^−3^) or OSCC (7.7/7.7 ± 2.5, *p* < 2 × 10^−18^) (Figure 4B). There was no significant difference between mild/moderate and severe dysplasias. Of all the dysplasias studied (*n* = 70), 24 were clinically diagnosed as oral leukoplakia (OL) which showed significantly higher qMIDS^V2^ scores (6.6/6.8 ± 2.6) to those of OSF (5.5/4.7 ± 2.8, *p* < 0.004; Figure 4C) and OLP (1.3/1.6 ± 1.3, *p* < 7 × 10^−9^; Figure 4C). The usually benign but inflammatory OLP (1.3/1.6 ± 1.3, Figure 4D) was not statistically different to NOM (1.4/1.8 ± 2.5, Figure 4B).

### 3.7. International Multi-Cohort Comparisons

Diagnostic performances for qMIDS^V2^ of Chinese (Figure 4A) and Indian (Figure 4B) cohorts were both similar to the UK (Figure 3) cohort confirming the robustness of the qMIDS assay against differences in tissue preservation methods (fresh frozen vs. FFPE), RNA purification methods (mRNA vs. total RNA), ethnicity and geographic regions (Figure 4E,F). qMIDS^V2^ assay sensitivity across the UK, China and India were 88%, 88% and 97%; assay specificity were 96%, 91 and 86%; assay accuracy were 92%, 89% and 93%, respectively (Figure 4F). Positive predictive values across UK, China and India were 96%, 96% and 92%; false positive rates were 4.5%, 9.1% and 14.3%, respectively. Negative predictive values were 88%, 77%, 94%; false negative rates were 12%, 13% and 3.3%, respectively. AUC for UK, China and India were 0.945, 0.928 and 0.932, respectively. Of note, the qMIDS^V2^ assay sensitivity appeared slightly higher in the Indian cohort (97%) compared to UK or China (both at 88%) at the expense of higher false positive rate in India (14.3%) and conversely lower false negative rate (3.3%) in the UK and China. These differences observed in Indian cohort could be attributed to the use of carefully selected FFPE archival samples compared with UK and China cohorts where fresh frozen tissues surplus to diagnosis were used.

### 3.8. Prediction of Malignant Transformation in Dysplasia

Of the 70 Indian dysplasia cases studied above (from Figure 4B), 30 patients had at least 5-year clinical outcome data. Apart from time to malignant transformation, none of the other clinicopathological features were statistically different between the non-transformed to OSCC and transformed to OSCC groups (Table 1). We found that qMIDS^V2^ significantly discriminated between the dysplasias that did not transform to malignancy (median/mean ± SD: 1.1/2.1 ± 2.4) compared to dysplasias that did transform to malignancy within 5 years (5.2/4.6 ± 2.4, Mann–Whitney U test *p* < 2 × 10^−6^; *t*-test *p* < 0.004; Figure 5A). Assessing the effectiveness of pathology dysplasia grading (WHO 2017 OED grading criteria reviewed in [11]), if severe dysplasia is a predictor of transformation (i.e., cut-off between mild/moderate and severe dysplasia), there were 4 severe dysplasias within the 15 dysplasias that did not transform compared to 9 out of 15 that transformed. This produced: sensitivity (60.0%), specificity (73.3%), accuracy (66.7%), false-positive rate (26.7%) and false-negative rate (40.0%) for dysplasia grading in predicting transformation (Figure 5B,C). Using a cut-off between mild and moderate/severe, improved sensitivity (86.7%) but reduced accuracy (56.7%) and very poor specificity (26.7%) (data not shown) were observed. Compared with using qMIDS^V2^ in predicting transformation, with a cut-off at 4.0 [24], within the 15 non-transformed dysplasias, 4 cases had qMIDS^V2^ index above 4.0, whereas within the transformed dysplasias, 4 cases had qMIDS^V2^ index below 4.0. This produced: sensitivity (73.3%), specificity (73.3%), accuracy (73.3%), false positive rate (26.7%) and false negative rate (26.7%). qMIDS^V2^ (at cut-off 4.0) could help reduce false-negative rate of dysplasia grading from 40% to 26.7%. If the cut-off values were lowered to 2.2 (an optimal cut-off level in this dataset), qMIDS^V2^ could further reduce false-negative rate to 13.3% while increasing test sensitivity from 60.0% to 86.7% and accuracy from 66.7% to 80.0% (Figure 5B,C).

## 4. Discussion

In 2013, we created and validated a multi-gene cancer diagnostic test, qMIDS^V1^ for OPMD and OSCC based on bioinformatics, cell culture and molecular selection techniques to identify key oncogenic driver genes [24]. The qMIDS^V1^ test was first validated on UK and Norwegian patient samples [24] and subsequently independently validated in China using ethnic Han Chinese specimens [30]. These initial tests were conducted on 1761 individual 1 mm^3^ tissue fragments collected from 427 samples from Caucasians and Asian patients which generated 24,654 gene expression data points from qMIDS^V1^. We exploited this dataset to identify and remove less influential genes and reformulated a second generation qMIDS^V2^ gene panel containing biomarkers representing tumour cells, abnormal matrix, blood vessels, and infiltrating immune cells to capture a more holistic picture of a tumour tissue. 

Among the 8 new genes, 5 of them (MMP13, INHBA, NR3C1, S100A16 and CXCL8/IL8) are known markers involved in stroma/matrix and immune modulation in OSCC. MMP13 (matrix metallopeptidase 13) had been shown to be expressed in the invading front of the tumour and in stromal fibroblasts [40] and its expression was significantly higher in large (>4 cm) locally invasive tumours [41]. INHBA (inhibin subunit beta A) was shown to be upregulated [42] and regulates lymph node metastasis in HNSCC [43]. NR3C1 (nuclear receptor subfamily 3 group C member 1) was implicated in HNSCC in a pan-cancer bioinformatics analysis involving 3000 tumours [44]. We have previously shown that low levels of S100A16 (S100 calcium binding protein A16) in OSCC significantly correlated with reduced 10-year overall survival and poor tumour differentiation [45]. Both OSCC tumour and serum expression of CXCL8/IL8 (C-X-C motif chemokine ligand 8/interleukin-8) were previously shown to be correlated with poor clinical outcome [46]. Salivary CXCL8 was shown to be a marker for OSCC [47] and ectopic expression of CXCL8 promoted cell proliferation and migration in OSCC cell lines [48]. The remaining 3 genes (CBX7, TOP2A and BIRC5) filled the gaps of tumour cell regulation in stem cell, epigenetic, genomic instability, proliferation and differentiation (see Figure 2A). CBX7 (chromobox 7) was used in an 11-gene signature to identify extra-capsular spread in OSCC patients without nodal metastases [49]. TOP2A (DNA topoisomerase II alpha) expression was demonstrated to be a prognostic marker for malignant conversion in head and neck dysplasia [50]. BIRC5 (baculoviral IAP repeat containing 5 or survivin), shown to be upregulated in many tumour types including HNSCC, is associated with DNA methyltransferases and multiple immune cells infiltration [51].

qMIDS^V2^ was first validated on a cohort of *n* = 282 OSCC (UK) specimens demonstrating significantly better diagnostic performance (21–26% increase) over qMIDS^V1^. Importantly, the false-positive rate was lowered from 29% to 4.5% and false negative rate was lowered from 28% to 12% (Figure 3C). Further international multi-cohort validation using OSCC samples from China and India (Figure 4), all demonstrated comparable diagnostic performance confirming that the qMIDS^V2^ assay was not affected by ethnicity, geographical differences or different tissue preservation (fresh frozen/RNALater and FFPE) and RNA extraction methods (magnetic bead vs. column filtration). Similar to our previous qMIDS^V1^ findings [24], OLP which is associated with inflammation but usually benign showed similar qMIDS^V2^ (index < 2.0) to those of normal oral mucosa, indicating that OLP is of low-risk and inflammatory status is not a confounding factor for qMIDS^V2^ assay.

This study provided further evidence that qMIDS could be used for risk stratification in patients with OPMDs and qMIDS^V2^ performed even better than qMIDS^V1^ especially when clinico-histopathological features did not correlate with disease outcome [4,6,52]. Our Indian cohort demonstrated that qMIDS^V2^ was able to differentiate between dysplasias that did not undergo malignant transformation within 5 years and dysplasias that did transform into OSCCs over that period. Furthermore, qMIDS^V2^ showed higher sensitivity (86.7% vs. 60%) and lower false-negative rates (13.3% vs. 40%) in predicting malignant transformation when compared to dysplasia grading (Figure 5). A false-negative test is of great importance in this context as it would mean a missed early treatment opportunity before the dysplasia transforms into cancer, with a consequently poorer patient outcome. Given the sensitivity and accuracy of the qMIDS^V2^ assay, we envisage that this may be a useful quantitative tool to identify high-risk OPMD lesions and to complement pathology. These findings suggest that combining qMIDS^V2^ analysis with histopathological grading of dysplasia should improve the diagnosis, prognosis and management of OPMDs compared with histopathology alone. Samples for qMIDS^V2^ can be collected simultaneous with the biopsy sample as qMIDS test only requires 1 mm^3^ of tissue posing no additional complexity or impact on the patient. In addition, multiple 1 mm^3^ samples can be taken in patients with wide field change to reduce sampling error.

We do recognise a major limitation of our Indian cohort involving a small equal number of non-transforming and transforming dysplasia cases, which is certainly not a true representative of the population [9,10]. However, despite a small cohort, qMIDS^V2^ in this study provided a pilot evidence that it has a potential to differentiate between OPMDs that undergo malignant transformation and those that do not.

There are other methods that are less invasive than tissue biopsy or have been developed to assist in risk assessments for OPMDs. Collecting saliva for tumour gene expression signature is less invasive but these cannot accurately locate where the tumour is and may not detect malignancies in deeper layers of the mucosal epithelium. Oral brush biopsy may be able to harvest sufficient tissue material directly from the oral lesion for qMIDS detection. This would render qMIDS non-invasive and could therefore be used for screening of OPMDs to pick up early malignant changes. In recent years, emerging new biofluid screening technology based on extracellular RNA (exRNA) biomarkers appears very promising [53,54]. Although there are many screening adjuncts in the market, none of them to date is able to identify high-risk from benign oral lesions with significant confidence [4,6,7,8,12,13]. A recent systematic review has indicated that none of the non-invasive adjunctive tests can be recommended as a replacement for histological assessment [55].

Following the pandemic of the Coronavirus disease 2019 (COVID-19), qPCR diagnostic infrastructure has been significantly up-scaled globally to cope with viral testing demand [56]. This would therefore enable significantly easier and more cost-effective integration of qPCR-based tests such as qMIDS to existing diagnostics infrastructure already running qPCR. As OPMDs are more prevalent in lower socioeconomic populations, improving accessibility to a cost-effective test such as qMIDS is pivotal to enable better case-finding in deprived and high-risk populations worldwide.

## 5. Conclusions

Collectively, these results demonstrated a holistic approach in capturing gene signatures from both the tumour cells and the tumour microenvironment and could significantly improve molecular diagnostic performance. This study provided an international multi-cohort validation of a second generation qMIDS^V2^ test on a total of *n* = 535 samples from UK (*n* = 282), China (*n* = 35) and India (*n* = 218) demonstrating a minimally invasive test that is robust against ethnobiological and geographical differences for OSCC detection. Despite in a small cohort, qMIDS^V2^ showed potential prognostic use for malignant transformation risk stratification in otherwise ambiguous dysplastic oral lesions. Further longitudinal long-term follow-up on prospective study on a larger cohort of OPMD patients is required to confirm the prognostic use of qMIDS^V2^ and to improve OPMD patient management and guide treatment decision.

## 6. Patents

Queen Mary University of London filled a patent application at the World Intellectual Property Organisation pertaining to the use of the qMIDS technology (biomarkers and algorithm) described in this paper for cancer diagnosis.

## Figures and Tables

**Figure 1 cancers-14-01389-f001:**
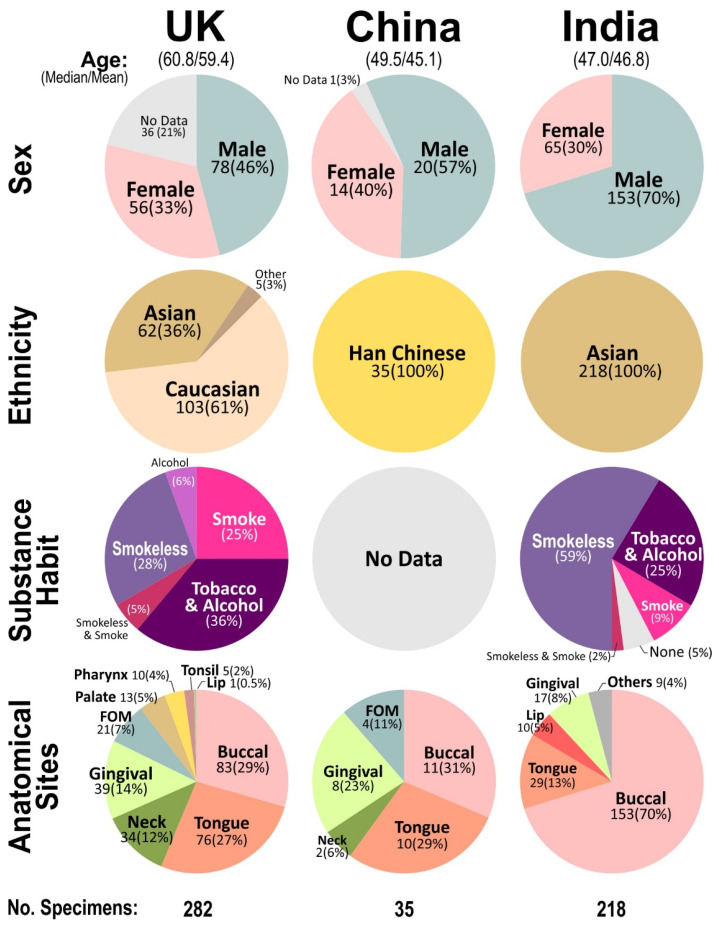
Summary of patient demographic data (age, sex, ethnicity, substance use habits and tissue anatomical sites) associated with all tissue samples used in this study from UK, China and India, respectively. The number of samples are indicated within the pie-chart with % in parenthesis under each data labels. In the UK cohort, each patient contributed either single or multiple samples (paired margins, core and neck metastasis), hence the total number of UK patients were 170 contributing 282 specimens. In China and India cohorts, each patient provided only a single tissue specimen.

**Figure 2 cancers-14-01389-f002:**
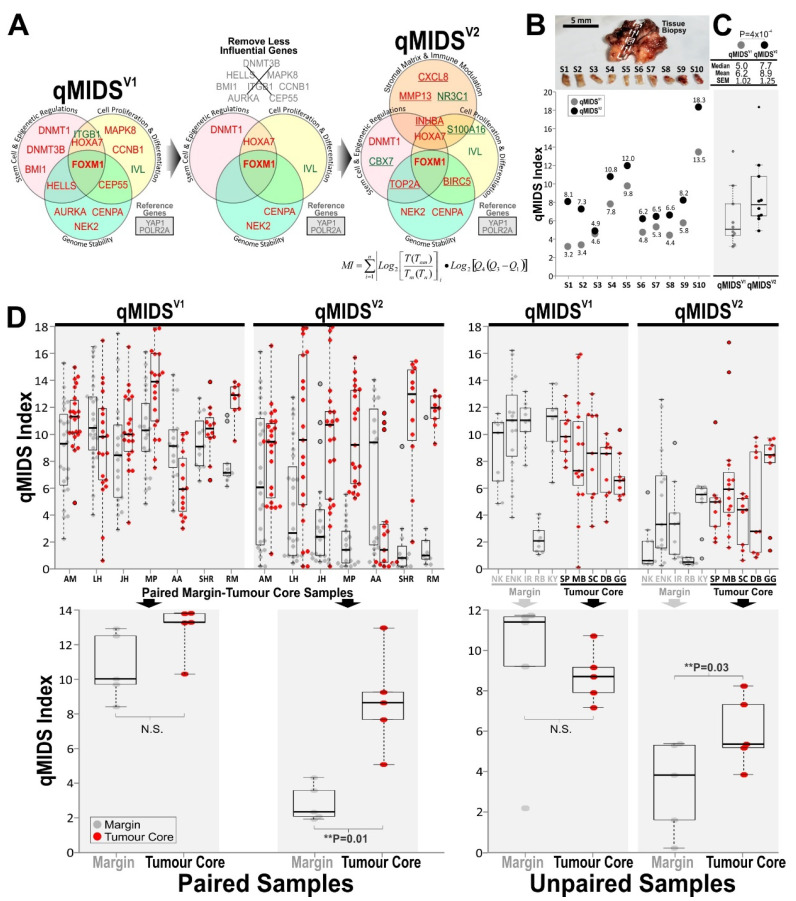
Comparisons between qMIDS^V1^ and qMIDS^V2^. (**A**) Biomarker gene panel and their respective functional groups in qMIDS^V1^ compared to qMIDS^V2^. Diagrams indicate the removal of less influential genes from qMIDS^V1^ and replacement of new genes with functional regulation of stroma matrix and immune modulation in qMIDS^V2^. The qMIDS algorithm [24] was used to compute 16 gene expression levels into a qMIDS malignancy index (MI) for each sample. (**B**), Case study using a single OSCC tumour core tissue biopsy for qMIDS^V1^ and qMIDS^V2^ comparison. Photograph showing the cut site of a strip of tissue across the tumour sample which was subsequently dissected into 10 pieces of 1 mm^3^ tissue fragments. Each fragment was subjected to qMIDS^V1^ and qMIDS^V2^ assays simultaneously and their corresponding qMIDS indexes were shown adjacent to the data points. Of note, some fragments (e.g., S1, S2 and S10) showed much larger differences between qMIDS^V1^ and qMIDS^V2^ which may reflect the molecular heterogeneity across the tumour tissue. (**C**), Data from (**B**) were plotted as box-whisker dot plots (box horizontal lines represent median and 25–75% percentiles, whiskers represent lowest and highest values, outliers are beyond the whiskers), *t*-test were performed. *p*-values were indicated in the panel above. (**D**), Similar to methods in (**B**,**C**), each sample was cut into 9–24 fragments for qMIDS^V1^ and qMIDS^V2^ comparison whereby paired and unpaired margin and tumour core samples were analysed. A total of *n* = 498 sub-fragments (from paired samples of 7 patients) and *n* = 204 sub-fragments (unpaired samples of 10 patients) were independently analysed. Top panels show box-whisker dot plots of individual fragments for each patient (*x*-axis showed individual patients’ sample IDs). Panels below show box-whisker dot plots of average values of all fragments from each sample and corresponding statistical *t*-test *p*-values.

**Figure 3 cancers-14-01389-f003:**
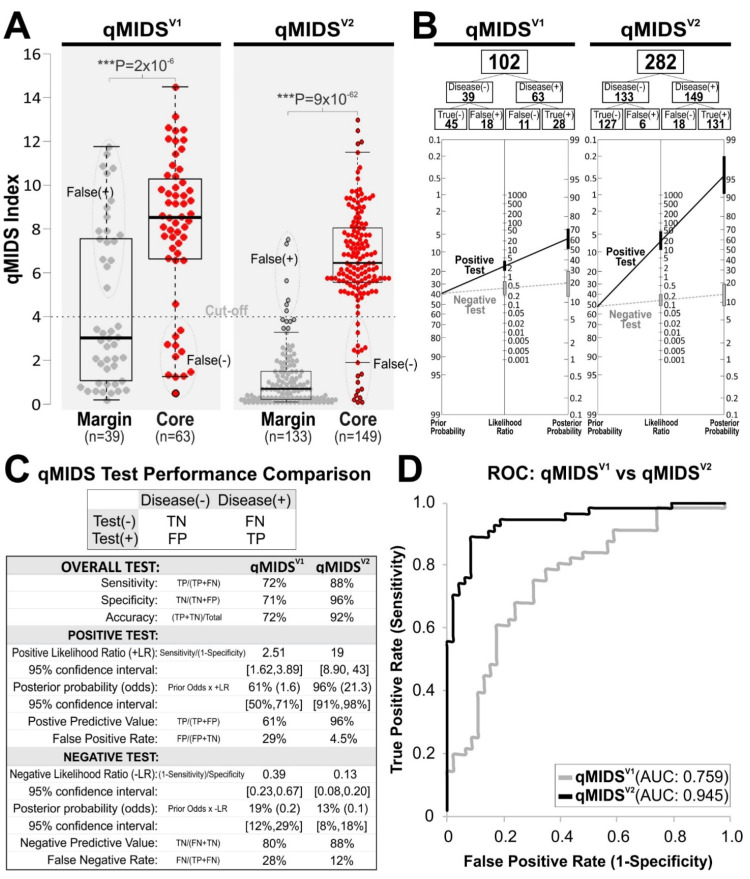
Independent validation in UK cohort on margin and OSCC tumour core samples showing data comparisons between qMIDS^V1^ and qMIDS^V2^. Due to insufficient tissue sample left for qMIDS^V1^ assays, only a subset of samples (*n* = 102) was analysed with qMIDS^V1^ compared to qMIDS^V2^ (*n* = 282). (**A**), Box-whisker dot plots (box horizontal lines represent median and 25–75% percentiles, whiskers represent lowest and highest values, outliers are beyond the whiskers) showing the segregation of data and *t*-test analysis *p*-values for qMIDS^V1^ and qMIDS^V2^. (**B**), Diagnostic test performances for qMIDS^V1^ and qMIDS^V2^ were calculated based on the cut-off value at 4.0 (dotted line as shown in (**A**). (**C**), Diagnostic test performance results for qMIDS^V1^ and qMIDS^V2^. TN, true negative; FN, false negative; FP, false positive; TP, true positive. (**D**), Data from (**A**) were separately subjected to ROC analysis showing the comparison between qMIDS^V1^ and qMIDS^V2^ with respective AUC values as shown within the panel.

**Figure 4 cancers-14-01389-f004:**
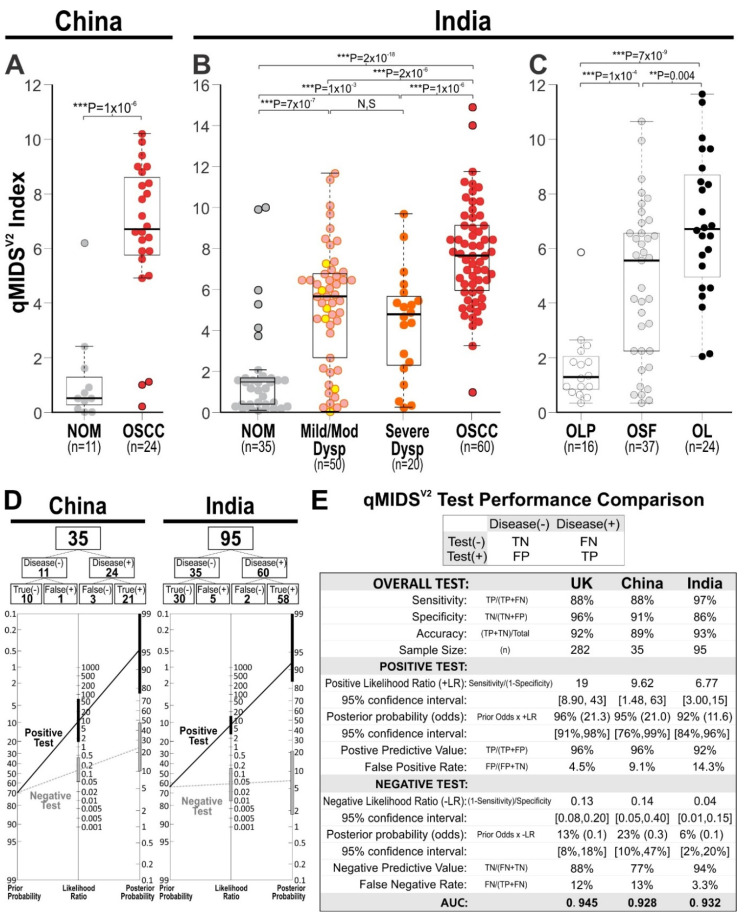
Multi-cohort qMIDS^V2^ diagnostic performance comparisons across geographically and ethnically distinct OSCC cohorts. (**A**,**B**), China cohort samples (snapped frozen samples): (**A**), normal oral mucosa (NOM; grey) and oral squamous cell carcinomas (OSCC; red). Student’s *t*-test *p*-value is indicated within the panel. (**B**,**C**), Indian cohort samples (FFPE): (**B**), Samples were grouped according to histopathology grading (WHO 2017): NOM, Mild (yellow)/Moderate (pink) Dysplasia (Dysp), Severe (orange) Dysplasia and OSCC (red). C, Oral lichen planus (OLP), submucous fibrosis (OSF) and dysplastic oral leukoplakia (OL) were compared. Outliers are indicated by black outlined symbols and *t*-test *p*-values are indicated above each chart. (**D**), Diagnostic test performance were compared between China and India OSCC cohort data obtained from (**A**,**B**). (**E**), Diagnostic test performance table for OSCC comparing between UK (extracted from Figure 3), China and India. TN, true negative; FN, false negative; FP, false positive; TP, true positive.

**Figure 5 cancers-14-01389-f005:**
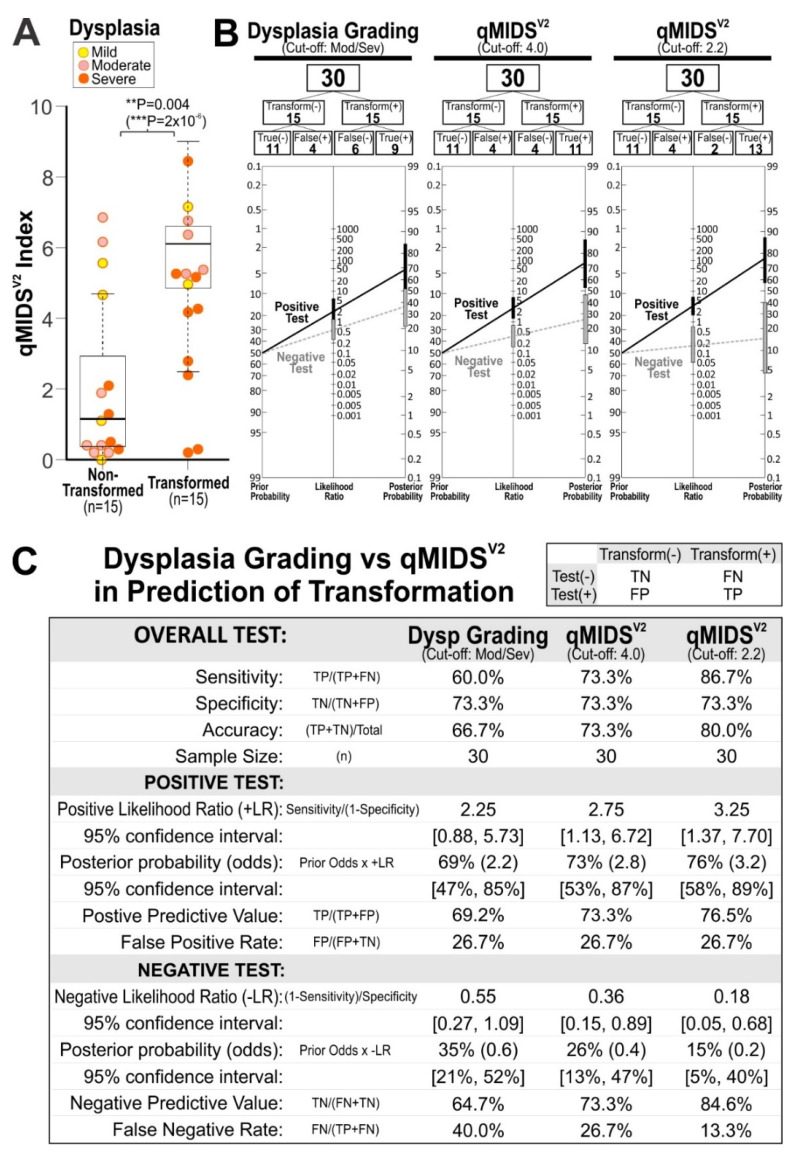
Comparison between pathological dysplasia grading method (WHO OED 2017) and qMIDS^V2^ for predicting malignant transformation in an Indian dysplasia cohort. (**A**), Dysplasia patients with 5-year outcome data (*n* = 30 from Figure 4B) were re-grouped according to their dysplasia transformation status: non-transformed or transformed into OSCC within 5 years. Student’s *t*-test *p* < 0.004 and Mann–Whitney U-test (*p* < 2 × 10^−6^) were performed due to skewed data distribution. Outliers are indicated by black outlined symbols and *t*-test *p*-values are indicated above the chart. (**B**), Prognostic performance of dysplasia grading (cut-off between mild/moderate and severe dysplasia grades) and qMIDS^V2^ (at cut-off 4.0 and 2.2) were analysed on the dysplasia cohort (*n* = 30) from (**A**) and their respective prognostic efficiencies are tabulated in (**C**). TN, true negative; FN, false negative; FP, false positive; TP, true positive.

**Table 1 cancers-14-01389-t001:** Clinicopathological features of dysplasia cases with 5-year clinical outcome.

Feature Category	Feature Subcategory	Non-Transformed	Transformed	*p*-Value
Age	Mean/Median	51.5/50.0	54.1/54.0	0.571 ^a^
	Standard Deviation	13.0	11.3	
Follow Up (months)	Mean/Median	38.4/36.0	6.7/1.9	<0.001 ^a^
	Interquartile Range	24	12	
Sex	Male	14	11	0.329 ^b^
	Female	1	4	
Site	Buccal Mucosa	10	9	1.000 ^b^
	Others	5	6	
Dysplasia,	Mild	4	2	
WHO 2017	Moderate	7	4	0.182 ^c^
	Severe	4	9	
Tobacco	Smoke-less	10	9	0.710 ^b^
	Smoking	5	6	

^a^ Student’s *t*-test; ^b^ Fisher’s exact test; ^c^ Pearson’s chi-square test.

## Data Availability

The data that support the findings of this study are available in the Appendix A of this article.

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
