# Peer review of "Molecular Signatures of Tumour and Its Microenvironment for Precise Quantitative Diagnosis of Oral Squamous Cell Carcinoma: An International Multi-Cohort Diagnostic Validation Study"

_cancers, 2022, doi:10.3390/cancers14061389_

Round 1
Reviewer 1 Report
This is a well-designed study to explore a rapid minimally invasive method with 14 biomarker genes for OSCC detection as well as prognostic use for malignant transformation risk stratification in dysplasia oral lesions. Authors developed an upgraded method based on their previous qMIDS method by replacing some less influential genes, and showed significant better diagnostic performance than previous method, which was further validated in several countries. This study provides very useful method in OSCC diagnosis and prediction of the malignant transformation of dysplasia lesions.
Suggestions:
- Please add more discussion about the detail functions and prognostic values of 8 new added biomarker genes in OSCC.
- Figure 5A showed that transformed dysplasia samples had significantly higher qMIDS index than non-transformed samples. However, some non-transformed samples showed relative high index (more than 4), while some transformed samples showed low index (less than 4). These phenomena have obvious effects on the prognostic accuracy of qMIDS v2 method in the prediction of malignant transformation in dysplasia. Please discuss the possible reasons and solutions for this problem.
Author Response
Reviewer 1:
This is a well-designed study to explore a rapid minimally invasive method with 14 biomarker genes for OSCC detection as well as prognostic use for malignant transformation risk stratification in dysplasia oral lesions. Authors developed an upgraded method based on their previous qMIDS method by replacing some less influential genes, and showed significant better diagnostic performance than previous method, which was further validated in several countries. This study provides very useful method in OSCC diagnosis and prediction of the malignant transformation of dysplasia lesions.
Suggestions:
- Please add more discussion about the detail functions and prognostic values of 8 new added biomarker genes in OSCC.
Our reply:
We thank the reviewer for the comments. As requested, we have now added a paragraph below detailing the functions of the 8 new biomarkers in relation to OSCC in the discussion section.
“Amongst the 8 new genes, 5 of them (MMP13 , INHBA, NR3C1, S100A16 and CXCL8/IL8) are known markers involved in stroma/matrix and immune modulation in OSCC. MMP13 (matrix metallopeptidase 13) had been shown to be expressed in the invading front the tumour and in stromal fibroblasts [40] and its expression significantly higher in large (>4 cm) locally invasive tumours [41]. INHBA (inhibin subunit beta A) was shown to be upregulated [42] and regulates lymph node metastasis in HNSCC [43]. NR3C1 (nuclear receptor subfamily 3 group C member 1) was implicated in HNSCC in a pan-cancer bioinformatics analysis involving 3000 tumours [44]. We have previously shown that low levels of S100A16 (S100 calcium binding protein A16) in OSCC significantly correlated with reduced 10-year overall survival and poor tumour differentiation [45]. Both OSCC tumour and serum expression of CXCL8/IL8 (C-X-C motif chemokine ligand 8/interleukin-8) were previously shown to be correlated with poor clinical outcome [46]. Salivary CXCL8 was shown to be a marker for OSCC [47] and ectopic expression of CXCL8 promoted cell proliferation and migration in OSCC cell lines [48]. The remaining 3 genes (CBX7, TOP2A and BIRC5) filled the gaps of tumour cell regulation in stem cell, epigenetic, genomic instability, proliferation and differentiation (see Figure 2A). CBX7 (chromobox 7) was used in an 11-gene signature to identify extra-capsular spread in OSCC patients without nodal metastases [49]. TOP2A (DNA topoisomerase II alpha) expression was demonstrated to be a prognostic marker for malignant conversion in head and neck dysplasia [50].“
- Figure 5A showed that transformed dysplasia samples had significantly higher qMIDS index than non-transformed samples. However, some non-transformed samples showed relative high index (more than 4), while some transformed samples showed low index (less than 4). These phenomena have obvious effects on the prognostic accuracy of qMIDS v2 method in the prediction of malignant transformation in dysplasia. Please discuss the possible reasons and solutions for this problem.
Our reply:
We appreciate the reviewer for the interesting observation. Figure 5A shows results of a small cohort (n=30) of dysplasia samples with 5-year follow up data. Due to difficulty of obtaining primary biopsy samples with a long follow-up clinical outcome history, we were unable to obtain more samples to be included in the study – this is our main limitation. Furthermore, the dysplasia samples were often very small (could have missed the main lesion) and extraction of sufficient quality RNA from these FFPE samples may distort qPCR results and hence the observed discrepancy. Sampling error and/or inclusion of large amount of normal oral mucosa could also be a confounding factor. Possible solutions would be to use fresh frozen biopsy and to obtain multiple minimally invasive 1 mm samples from different areas of the oral lesion in each patient to reduce sampling error.
Reviewer 2 Report
OPMD are associated with a statistically increased risk of OSCC. In terms of primary prevention their early detection is relevant in head and neck oncology worldwide. On the other hand, it is difficult to predict the development of oral cancer. Therefore, adequate tests are urgently needed regarding high precision at a reasonable small sample size. I fully support the author’s approach.
The authors developed a good validation strategy for the mentioned biomarker panels and based on their previous research.
The cohort and sample sizes are large and contain different ethnicities. Although the Chinese cohort is really small and the clinical data are incomplete.
I suggest some minor modifications:
Preanalytic sample handling is very inconsistent. RNA isolation from FFPE vs. RNALater results in substantial differences in RNA quality and subsequently PCR results. Please comment on that.
In general, the figures are no vector graphics and the resolution is too low. The authors have to correct.
The figures appear overloaded. I understand there is al lot of information to carry but maybe a worldmap with three dots representing three countries is not necessary.
The density of information varies too much. Please compare figure 2D with the pie charts of china row in figure 1. Please increase readability.
Again, the Chinese cohort is very small. Maybe there’s a chance to figure out the noxious habits?
What about HPV infection and detection via PCR? Please comment on that.
Table 1
The label „WHO 2017“ should be changed to „Dysplasia, WHO 2017“
Figure 1
All samples are taken from mucosa. What`s the meaning of "gingival", "mandible" and "maxilla"? It should be redefined in „lower alveolus gingiva“ or „upper alveolus gingiva“.
How was it assured that the samples from neck metastases are really from histological verified neck metastases? Was is it just the macroscopic aspect?
Tonsils are part of the oropharynx, but the title of the manuscript is about OSCC. Please clarify.
Formal:
Line 282 and 300: Please superscript “3”
Once an abbreviation like OLP is introduced, there is no need to repeat the explanation
Author Response
Reviewer 2:
OPMD are associated with a statistically increased risk of OSCC. In terms of primary prevention their early detection is relevant in head and neck oncology worldwide. On the other hand, it is difficult to predict the development of oral cancer. Therefore, adequate tests are urgently needed regarding high precision at a reasonable small sample size. I fully support the author’s approach. The authors developed a good validation strategy for the mentioned biomarker panels and based on their previous research. The cohort and sample sizes are large and contain different ethnicities. Although the Chinese cohort is really small and the clinical data are incomplete.
I suggest some minor modifications:
Preanalytic sample handling is very inconsistent. RNA isolation from FFPE vs. RNALater results in substantial differences in RNA quality and subsequently PCR results. Please comment on that.
Our reply:
Thank you for this important comment. It is indeed very challenging to work with FFPE samples to obtain good quality RNA from samples which are 6-12 years old and furthermore some samples (especially the dysplasias) were very small. We spent a number of years optimising our method in order to obtain good quality qPCR results from FFPE samples. This involved using one-step RT-qPCR whereby we elute each RNA sample in very small volume of water (15 µL) and the whole eluted RNA was immediately used directly for reverse transcription and qPCR using gene-specific primers in the same reaction. This significantly improved qPCR results over the conventional 2-step method. Furthermore, we exclude samples that do not meet our minimum inclusion criteria whereby both reference genes should be detectable below ct of 35 cycles. In an ideal situation, using fresh frozen sample could mitigate against the low-quality RNA issue.
In general, the figures are no vector graphics and the resolution is too low. The authors have to correct.
Our reply:
We have now fixed the resolution of the figures.
The figures appear overloaded. I understand there is al lot of information to carry but maybe a worldmap with three dots representing three countries is not necessary.
Our reply:
Thank you for the suggestion. The worldmap is actually part of the Graphical Abstract which is necessary to illustrate the international multi-cohort study. As suggested, this has now been made smaller and fonts of the figure has been made bigger to make it more readable.
The density of information varies too much. Please compare figure 2D with the pie charts of china row in figure 1. Please increase readability.
Our reply:
We have now eidited the figure to make Figure 2D bigger for better readability.
Again, the Chinese cohort is very small. Maybe there’s a chance to figure out the noxious habits?
Our reply:
We appreciate the reviewer’s concern about the Chinese cohort. Due to time and funding limitations, we were not able to obtain more samples. Also, oral cancer cases in our collaborating Chinese hospital were not as high as expected. According to our previous qMIDSV1 validation study in a larger Chinese cohort with n=68 patients (Ma et al., Oncotarget. 2016 Aug 23;7(34):54555-54563. doi: 10.18632/oncotarget.10512.), noxious habits were largely due to smoking and drinking. Unfortunately, this was anecdotal evidence therefore we did not include habits information in the paper for the Chinese cohort.
What about HPV infection and detection via PCR? Please comment on that.
Our reply:
We did not investigate HPV infection in our study because 99% of our samples are from the oral cavity, hence it is beyond the scope of our study to detect HPV. HPV mostly concerns oropharyngeal SCC rather than oral cavity SCC. Nevertheless, it would be interesting to perform HPV detection in a separate study. It is worthy to note that significant set of experiments would be required to profile the various different HPV types and also to differentiate between episomal vs integrated HPV in each patient samples.
Table 1
The label „WHO 2017“ should be changed to „Dysplasia, WHO 2017“
Our reply:
We have now added this label as suggested.
Figure 1 All samples are taken from mucosa. What`s the meaning of "gingival", "mandible" and "maxilla"? It should be redefined in „lower alveolus gingiva“ or „upper alveolus gingiva“.
Our reply:
Thank you for the suggestion, we have now edited accordingly.
How was it assured that the samples from neck metastases are really from histological verified neck metastases? Was is it just the macroscopic aspect?
Our reply:
All samples were classified according to histopathological findings and verified by our collaborating oral pathologists.
Tonsils are part of the oropharynx, but the title of the manuscript is about OSCC. Please clarify.
Our reply:
Thank you for pointing out this issue. Because the majority (n=530) of our samples are from the oral cavity, only n=5 were tonsils, we felt that the OSCC term is more representative to reflect the 99% of samples compared to the term HNSCC.
Formal:
Line 282 and 300: Please superscript “3”
Once an abbreviation like OLP is introduced, there is no need to repeat the explanation
Our reply:
Thank you for identifying these mistakes, we have now corrected them in the revised manuscript.